# Sensorimotor mechanisms selective to numerosity derived from individual differences

**Giovanni Anobile[1]\*, Irene Petrizzo[1], Daisy Paiardini[1], David Burr[1,2], Guido Marco Cicchini[3]**

[1]Department of Neuroscience, Psychology, Pharmacology and Child Health, University of Florence, Florence, Italy; [2]School of Psychology, University of Sydney, Camperdown NSW, Sydney, Australia; [3]Institute of Neuroscience, CNR, Pisa, Italy

**Abstract** We have previously shown that after few seconds of adaptation by finger-tapping, the perceived numerosity of spatial arrays and temporal sequences of visual objects displayed near the tapping region is increased or decreased, implying the existence of a *sensorimotor numerosity system* (Anobile et al., 2016). To date, this mechanism has been evidenced only by adaptation. Here, we extend our finding by leveraging on a well-established covariance technique, used to unveil and characterize 'channels' for basic visual features such as colour, motion, contrast, and spatial frequency. Participants were required to press rapidly a key a specific number of times, without counting. We then correlated the precision of reproduction for various target number presses between participants. The results showed high positive correlations for nearby target numbers, scaling down with numerical distance, implying tuning selectivity. Factor analysis identified two factors, one for low and the other for higher numbers. Principal component analysis revealed two bell-shaped covariance channels, peaking at different numerical values. Two control experiments ruled out the role of non-numerical strategies based on tapping frequency and response duration. These results reinforce our previous reports based on adaptation, and further suggest the existence of at least two sensorimotor number channels responsible for translating symbolic numbers into action sequences.

## eLife assessment

This potentially **important** paper addresses the question of how numerical information is represented in the human brain. Experimental findings are interpreted as providing evidence for a sensorimotor mechanism that involves channels, each tuned to a particular numerical range. While this is an interesting application of methodologies used to identify the presence of channels, the evidence supporting the claim that these have a sensorimotor basis is **incomplete**.

## Introduction

Humans share with many animals a primitive non-verbal number system encoding the number of objects in space and events in time (*Dehaene, 2011*). The capacity to estimate number rapidly but imperfectly is thought to be a primary perceptual attribute, often termed the *number sense* (*Burr and Ross, 2008*). The past few decades have accumulated much evidence for the existence of the number sense. Many studies show that the sense of number is truly general, transcending space and time, as well as modality and format (*Arrighi et al., 2014*; *Burr et al., 2017*; *Togoli et al., 2021*). More recently, numerosity has been shown to interact strongly with action, leading to the idea of a

\*For correspondence:
giovanni.anobile@unifi.it

**Competing interest:** The authors declare that no competing interests exist.

*sensorimotor number system*, incorporating action, and its interaction with sensory systems (*Anobile et al., 2016*; *Anobile et al., 2021*).

Both physiological and psychophysical studies support the existence of a sensorimotor number system. Adaptation studies provide strong evidence: participants first tap continuously with their index finger for a few seconds, either rapidly or slowly, then estimate the numerosity of a dot array presented near the adapted region: adaptation to fast tapping causes underestimation and slow tapping overestimation (*Anobile et al., 2016*; *Anobile et al., 2020*; *Maldonado Moscoso et al., 2020*). Motor adaptation induces similar effects on sequential visual and auditory stimuli (*Anobile et al., 2016*; *Togoli et al., 2020*), consistent with the existence of a generalized system linking motor and sensory signals to encode numerosity (*Anobile et al., 2021*; *Burr et al., 2021*).

Electrophysiological studies in primates also point to the existence of a specific neural substrate for counting a small set of actions. In a seminal paper, *Sawamura et al., 2002* trained monkeys to repetitively make five identical movements, then switch to a different movement, in a cyclical fashion: neurons in the posterior parietal cortex showed selectivity to the number of self-generated actions, whatever the action. *Kirschhock and Nieder, 2022* trained crows to peck a specific number of times (1–5). Neurons in the telencephalon were shown to be tuned to the impending number of self-generated actions, during the phase between the disappearance of the target to the onset of motor reproduction. The activity of these neurons predicted the behavioral performance and was independent of both stimulus format (dots or digits) and of the temporal characteristics of the motor responses. Each tuning function peaked at a given preferred numerosity, with activity scaling down with numerical distance. Overall, these cells in the crow brain could constitute the neural substrate subserving the transformation of sensory inputs into a given quantity of numerical actions, possibly a similar mechanism to that driving the motor number adaptation effects in humans (*Anobile et al., 2016*).

Although motor adaptation has revealed a clear link between action and numerosity perception, this is the only technique used so far to investigate the interaction in humans. The aim of the current study is therefore to expand and generalize our previous findings, using a different psychophysical technique that exploits individual differences in reproduction precision (*Peterzell and Kennedy, 2016*). This technique has been widely used to reveal visual channels for motion (*Morrone et al., 1999*), spatial frequency (*Reynaud and Hess, 2017*; *Simpson and McFadden, 2005*), contrast sensitivity (*Peterzell et al., 1995*; *Peterzell and Teller, 1996*), color (*Peterzell et al., 2000*; *Peterzell and Teller, 2000*), and duration (*Rammsayer and Troche, 2014*).

The rationale behind this technique is that performance measures of stimuli detected by the same mechanism should correlate more between individuals than stimuli detected by different mechanisms. *Figure 1* illustrates the technique by simulation of results in a numerosity reproduction task if the task were mediated for 1, 2, or 4 numerosity-selective channels. It assumes intrinsic variability in reproduction precision across participants, perturbed by additive random noise. The top row simulates results if reproduction of all numerosities were governed by a single mechanism, rather than a range of numerosity-tuned mechanisms. The correlation matrix for all pairs of stimuli (*Figure 1B*) shows no systematic pattern, only random variations in correlations due to the general noisiness of the channel. *Figure 1C* plots the average correlation plotted as a function of numerosity ratio, showing no dependence on number ratio. Increasing or decreasing the added noise will change the average correlation, but not create any dependency on numerosity ratio. However, if there exist mechanisms selective for numerosity, the results are quite different. Assuming just two mechanisms tuned to low and high numbers (*Figure 1D*) yields a correlation matrix with higher correlations between similar numbers (near the diagonal) than dissimilar numbers. This leads to the clear dependency on numerosity ratio shown in *Figure 1E*. Similarly, for a range of numerosity-selective mechanisms like the four shown in *Figure 1G*, there will be a strong dependency on numerical distance (*Figure 1I*). The cases of two and four channels are difficult to distinguish if the amount of added noise is free to vary. Techniques such as cluster analysis and principal component analysis (PCA) can be also applied to study further the tuning of the channels.

Here, we applied the interindividual covariance technique to study sensorimotor tuning in humans, using a number matching task similar to that used by *Kirschhock and Nieder, 2022*. We measured the precision with which 30 participants could press a key a given number of times (8-32) without counting, and then correlated performance across all pairs of numbers (like the simulation of *Figure 1*). The

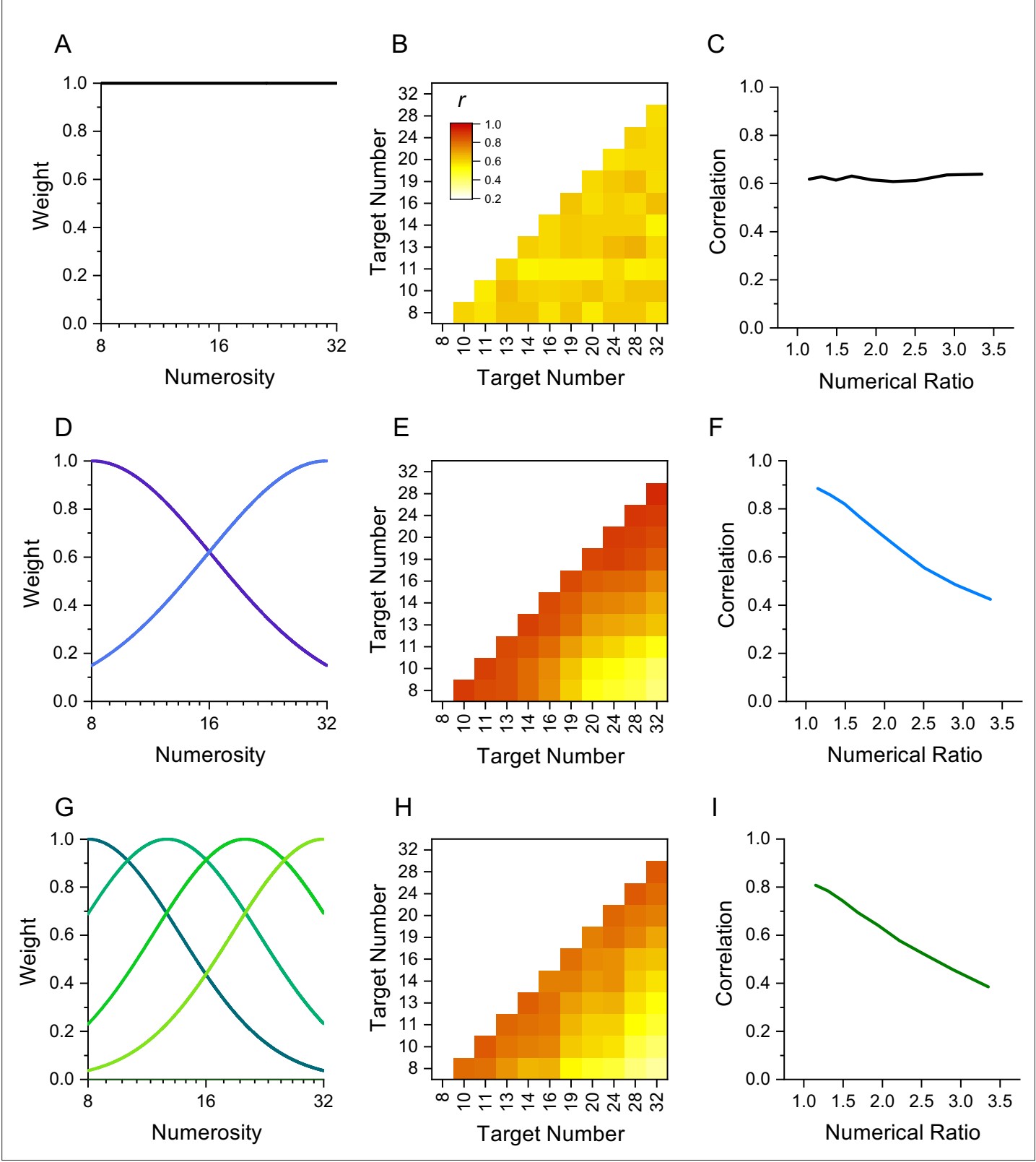

**Figure 1.** Simulation of behavior with 1, 2, or 4 channels mediating numerosity judgments. (**A**) 1-channel model shown as a flat tuning function. (**B**) Predicted correlation matrix between all pairs of numerosities. There is no systematic variability, only random variations induced by the noise. (**C**) Average correlation as function of numerical ratio of dot pairs, showing no dependency. (**D**) Two-channel model centred at numerosities 8 and 32, with full bandwidth of 1.45 octaves. (**E**) Correlation matrix, showing higher correlations near the identity line. (**F**) Correlation as function of numerical

*Figure 1 continued on next page*

*Figure 1 continued*

ratio, showing a systematic falloff with numerical ratio. (**G–I**) Same as D-F for a four-channel model with channels equally spaced in the range between 8 and 32 with full bandwidth of 1.10 octaves. Correlation matrices were obtained by simulating 1000 experiments with cohorts of 29–30 observers to yield the best fit ($R^2$).

---

results clearly demonstrate the existence of sensorimotor mechanisms converting symbolic numbers into actions, with correlations decreasing with numerical distance, like *Figure 1F, I*.

## Methods

### Participants

An a priori power analysis for a correlation test (r) with a medium effect size of $r^2=0.5$, and $\alpha=0.05$ (one tailed, as positive correlations are expected) and power of 0.9 indicated a required sample size of 28 participants. Thirty participants took part in the fast-tapping condition (age: average = 25.75, SD = 4.5, min = 18, max = 39) and 29 in the slow tapping condition (age: average = 25.43, SD = 4.04, min = 18, max = 39). Of these, 18 completed both conditions. The experimental procedures were approved by the local ethics committee (*Commissione per l'Etica della Ricerca*, University of Florence, July 7, 2020, n. 111). The research was in accordance with the Declaration of Helsinki and informed consent were obtained from all participants prior to the experiment.

### Stimuli and procedures

Stimuli were generated and presented with PsychToolbox routines for Matlab (ver. R2021a). Stimuli were white visual digit numbers (8, 10, 11, 13, 14, 16, 19, 21, 24, 28, 32) presented in the centre of a grey screen (iMac Retina display 27–inch) for 1 s and subtending 5° of visual angle. Soon after the disappearance of the target digit, participants were asked to repeatedly press a key as many times the target. In separate sessions, participants were asked to tap as fast as they could or at a comfortable rate. Participants were all right-handed and performed the tapping's with their right hand on a spacebar positioned about 30 cm to the right of the monitor, to guarantee a comfortable arm position. Following previous studies (*Cordes et al., 2001*; *Whalen et al., 1999*), serial counting was prevented by vocal suppression, repeating aloud the syllable 'ba' (as fast as possible). For each participant, each target number was presented from 25 to 30 times (for a total of 17,183 trials: 8259 and 8924 trials in the fast and slow tapping conditions), in separate blocks (usually 5) interspaced by pauses of a few minutes. For those participants who completed both the fast and slow tapping condition, the two were measured on different days (on average within 3–4 days). Each condition took about 1.5 hr of testing (3 hr for those who completed both). Before the experiment, participants were familiarized

---

**Table 1.** Responses duration (in seconds) tested in in the control duration task for each one of the nine participants (S1-S9).

| S1 | S2 | S3 | S4 | S5 | S6 | S7 | S8 | S9 |
|------|------|------|------|------|------|------|------|------|
| 3 | 4.1 | 7.3 | 3.5 | 2.2 | 5 | 2.4 | 2.6 | 2.3 |
| 4 | 6.3 | 9.2 | 4.9 | 3.8 | 6.3 | 3.3 | 2.8 | 2.9 |
| 4.4 | 7 | 10.5 | 5.4 | 4.2 | 6.5 | 3.5 | 3 | 3.2 |
| 5.1 | 7.9 | 11.3 | 6.2 | 5 | 8.3 | 4.2 | 3.8 | 4 |
| 5.7 | 9.3 | 12.5 | 7 | 5.4 | 8.6 | 4.3 | 4 | 4.4 |
| 6.8 | 10 | 15.3 | 8.2 | 6 | 9.5 | 5.4 | 4.4 | 5 |
| 7.8 | 11.7 | 17 | 9.9 | 6.9 | 11.5 | 6.4 | 5.3 | 5.8 |
| 8.8 | 13.7 | 17.4 | 10.6 | 7 | 13.2 | 7.4 | 5.7 | 6 |
| 10.2 | 15.8 | 21 | 12.2 | 8.6 | 15.3 | 8.1 | 6 | 7.4 |
| 12 | 17.8 | 23.3 | 13.3 | 9.2 | 17.7 | 9.6 | 7.2 | 8.8 |
| 12.9 | 21.7 | 29.2 | 16 | 10.8 | 20.6 | 11.1 | 8.2 | 9.4 |

with the task performing one single block of trials with feedback (a digit displaying the number of tappings performed). In this phase, 11 trials were presented, one for each tested number (randomly selected trial by trial). No feedback was provided during the rest of the experiment.

## Control duration task

A randomly selected sub-sample of nine participants from the slow tapping condition was engaged in a duration control task. On each trial, a tone (500 Hz, ramped on and off with 20ms raised-cosine ramps) was played though headphones, and participants asked to reproduce its duration by repeatedly tapping on a key while pronouncing the syllable 'ba' as fast as possible (to suppress counting). The between average tapping frequency rate in the number task was 3.1 Hz (SD 1.1 Hz) and 3.5 Hz (SD 1.3 Hz) in the duration task ($t_{(8)}$=1.03, p=0.33). Target durations were customised for each participant to perfectly match those previously produced in the number task: for each target number tested in the number matching task, we extracted the average response duration (across trials) and these eleven durations were re-presented as duration targets (*Table 1*). Before the experiment, participants were familiarized with the task by performing one single block of trials with feedback (2 numbers, one displaying the target duration and the second displaying the reproduced duration, in seconds). In this phase, 11 trials were presented, one for each tested duration (randomly selected trial by trial). No feedbacks were provided during the rest of the experiment. As for the number task, for each participant, each duration was presented from 25 to 30 times (for a total of 2455 trials), in separate blocks.

## Data analyses

All the analyses were performed with Matlab software (ver. R2021a) with the exception of the repeated measures ANOVA comparing Weber fractions for the number and duration task and the t-test comparing frequency rates for the number and duration task (performed with JASP software, ver. 0.16.3).

## Data processing

As a first step, we detected and eliminated outlier responses. For each experimental condition, and separately for each participant and stimulus intensity, responses were converted into z−scores and eliminated from the analyses if falling above or below 3 STD. In the number matching task, this procedure resulted in the elimination of 0.5% of the trials in the fast-tapping condition and 0.6% in the slow tapping condition. In the control duration task, the same procedure led to the elimination of 0.4% trials in the duration matching condition and 0.6% in the number matching condition.

As a second step, we measured responses accuracy and precision, on non-standardized data. For each task, and separately for each participant and stimulus intensity (numbers or durations), accuracy was indexed as the mean reproduction value across trials and precision as Weber fraction, computed as responses standard deviation divided by the average reproduction (a standard index of normalized precision).

## Correlation matrices and numerical distance

As a third step, we computed Weber fraction correlation matrices. We first measured the precision in translating digit numbers into action sequences, for different target numbers (8, 10, 11, 13, 14, 16, 19, 21, 24, 28, 32), for all N participants. For each target number, and independently for each participant, we calculated the normalized reproduction precision (Weber Fraction). The dataset comprised a matrix where each column represents a participant, each row a target number, and each cell the corresponding Weber Fraction. This dataset was then analysed with a pairwise correlation, across participants. For example, the WFs provided by the N participants when tested at the target number '8' were correlated with those obtained with the target number 10, 11, 13...32. If channels exist, the correlation between '8 and 10' (low numerical distance) should be higher compared to that between '8 with 32' (higher numerical distance). We therefore analysed the correlation strength as a function of numerical distance. This last analysis (*Figure 2*, panels B, D) was performed on binned data. Bins were created to have an approximately equal number of observations for each bin (13, 13, 14, 15). For each target, the numerical distance was calculated as the base ten logarithm ratio between itself and the remaining target numbers. The correlation coefficients were then averaged within the following numerical distance bins (log10 ratios): <0.08; 0.08–0.14; 0.14–0.21; 0.21–0.29, 0.29–0.39, >0.39.

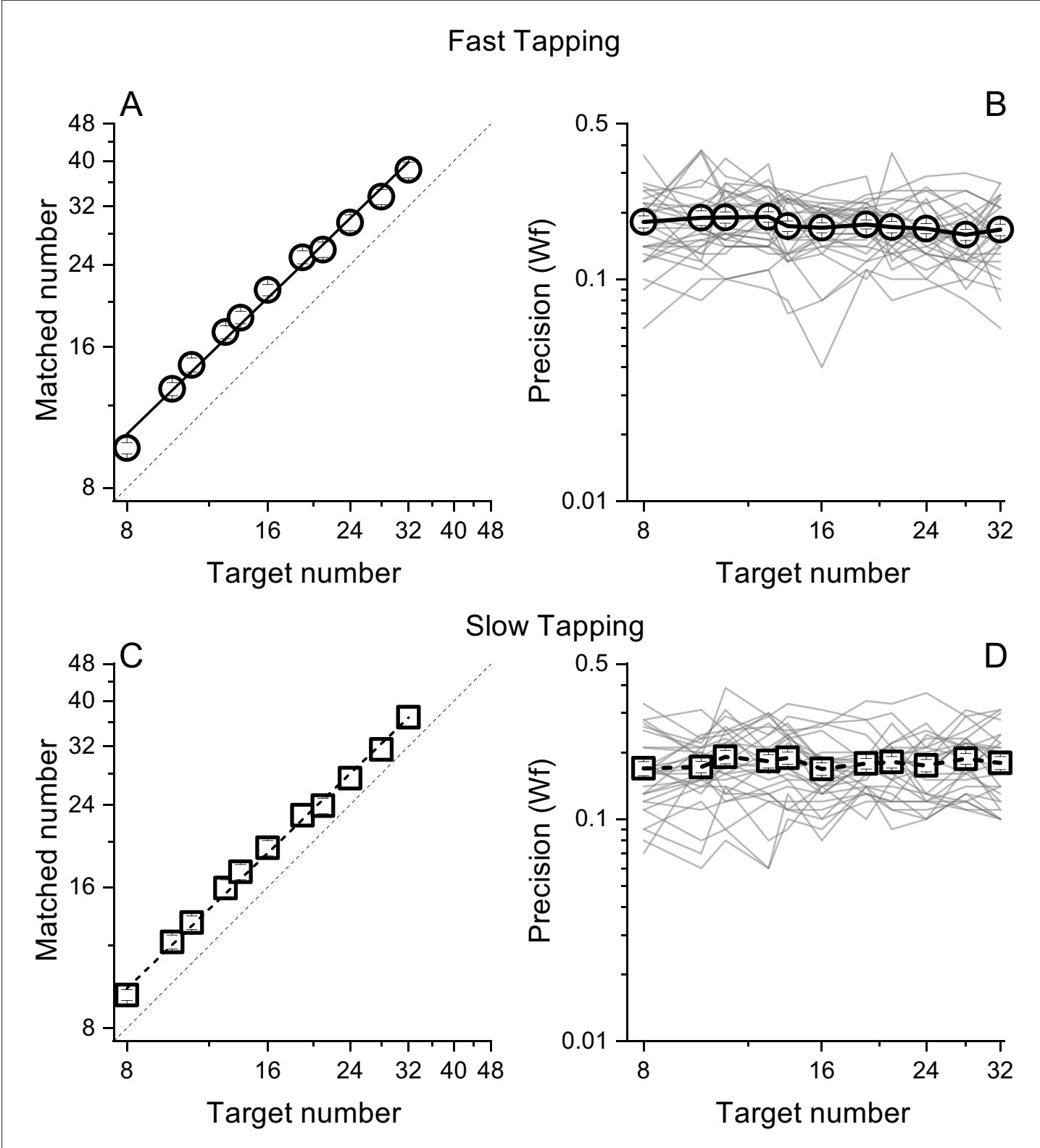

**Figure 2.** Matched number (A and B) and Weber Fractions (B and &C) as a function of target number, separated for the fast (circles) and slow (squares) tapping conditions. Lines through the data in panels A and C show best linear fits. Thin lines in panel B&D show individual data. Symbols represent means (N30 and N29 for fast and slow tapping respectively), Error bars are ±1 s.e.m., across participants.

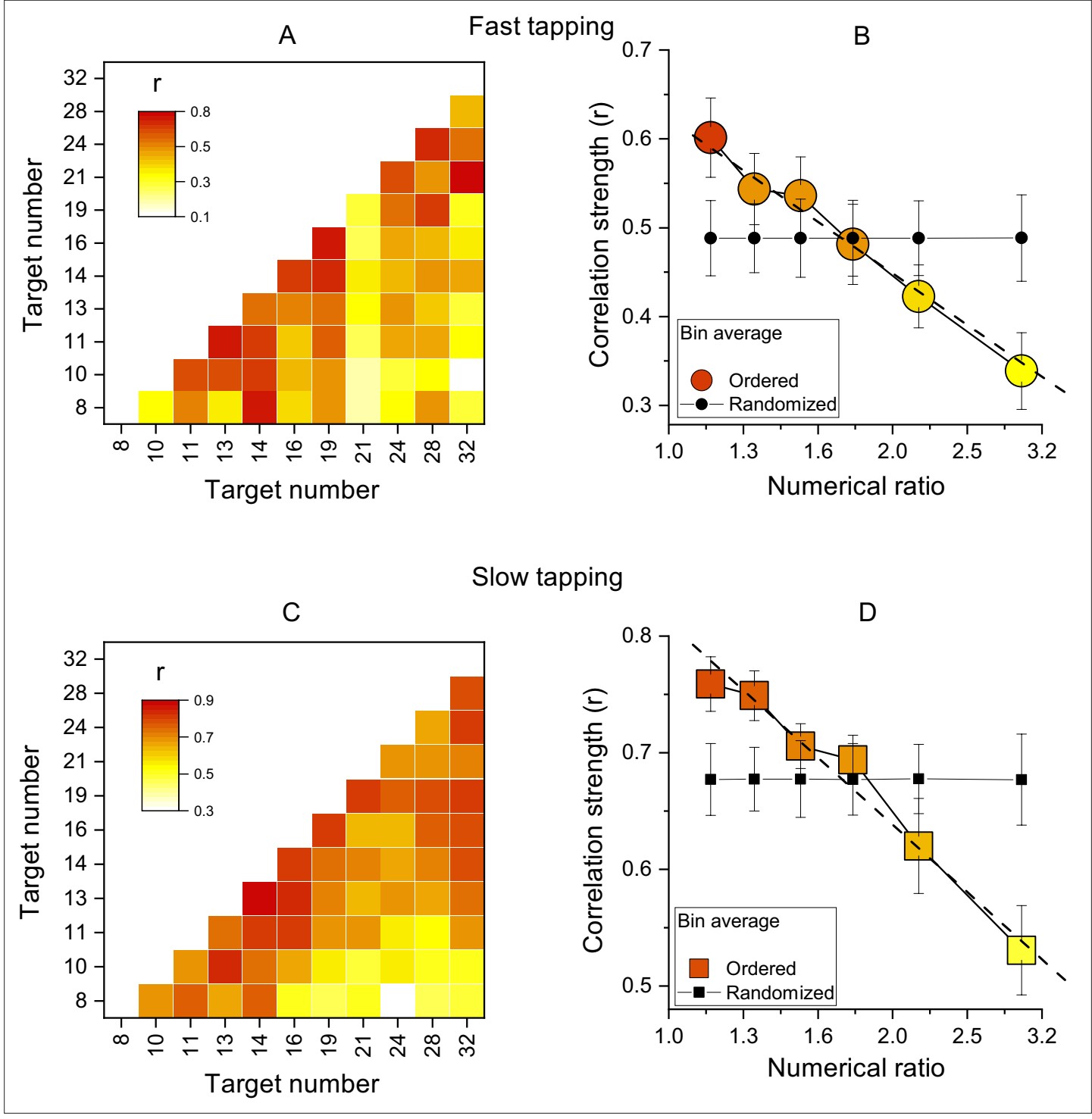

**Figure 3.** Inter-participant correlations. (A) Correlation matrices of Wfs for all pairs of target numbers for the slow-tapping condition. (B) Correlation strength as a function of numerical ratio, for the slow-tapping condition. Small black filled symbols show bootstrapped average correlation strengths of randomized Wfs matrices. Dashed lines are the best linear fit on unbinned data and error bars are ±1 s.e.m. Symbols represent means (N30 and N29 for fast and slow tapping respectively), best linear fit slopes were: –0.58±0.11 and –0.57±0.07 for fast and slow respectively, both different from zero p<0.001. C&D. Like A and B, for the slow-tapping condition.

As a sanity check, this analysis was also performed on randomized correlation matrices. For this analysis we applied a bootstrap approach. On each of 10,000 iterations, separately for each subject, the Weber fractions were randomized between numerosities and a new correlation matrix computed. Then (as before) the correlation coefficients were averaged within the reported bins. The results of this analysis are reported in *Figure 3* (panels B and D) as the random' curve.

### Hierarchical clustering and principal component analysis

The hierarchical clustering was performed on normalised (z-scores) Weber fractions using the linkage method. The distances were calculated with the Euclidean metric and the 'ward' algorithm was used to compute the distance between clusters. The number of clusters was determined by the inconsistency index (computed by the build in *inconsistent* function in Matlab 2021a) and results reported as a dendrogram. As for the PCA, to check the validity of this analysis, we tested it with randomized correlation matrices, with a bootstrap approach. On each of 10,000 iterations, separately for each participant, the Weber fractions were randomized between numerosities, and a new clustering was computed. Then at each iteration, we looked at the structure of the clusters, counting the time on which the clusters (separately for the fast and slow tapping conditions) contained contiguous numbers (as in *Figure 4A, B*).

The principal component analysis was performed on normalised (z-scores) and centred Weber fractions and factors rotated with the non-orthogonal promax method (but orthogonal varimax provides similar results). The number of components was determined as those exceeding an eigenvalue of 1 and from visual inspection of the scree plots.

### Modeling

As a preliminary analysis, we modeled the behavior of a system with 1, 2 or 4 channels, assuming that each channel would process stimuli with a gaussian tuning (like the illustration of *Figure 1*). The one-channel model had infinitely broad tuning, and the tuning of the two- and four-channel models was allowed to vary between 0.5 and 2octaves. The peak of the tuning was equally spaced in log coordinates to be 8 and 32 for the two-channel model and 8, 12.7, 20.1 and 32 for the four-channel model. The behavior for numerosities handled by more than one channel was determined by a weighted average of the various channels. The average WF of the channels taken from the actual dataset (i.e. 0.176 for the slow and fast tapping); however, the specific values for each observers could vary following a gaussian distribution whose standard deviation was varied to achieve best fit. We then calculated predicted correlations simulating 25 repetitions for each numerosity and observer. Without any noise correlations between the same channel would approximate 1. Hence to achieve reasonable levels of correlation, we assumed that the actual measured WF was corrupted by some noise which we manipulated to mimic the average correlation between observed in the actual experiment. Correlation matrices were obtained by simulating 1000 experiments with cohorts of 29–30 observers to yield the best fit ($R^2$).

## Results

Participants pressed a key a specific number of times (visually displayed as a digit) while repeatedly pronouncing the syllable 'ba' as fast as possible, to prevent counting. To gauge the generalizability across different motor actions, in separate sessions, participants were required to tap concurrently as fast as they could (fast tapping) or at a comfortable rate (slow tapping). As expected, the rate of tapping was higher in the fast tapping (average = 6.8 Hz, SD = 1) compared with the slow tapping condition (average = 3.3 Hz, SD = 1.6). In both conditions, participants were able to correctly perform the numerical matching task, with average number of reproduced actions scaling linearly with visual targets (both *r*=0.99, *Figure 2A, C*). *Figure 2B, D* shows precision (Weber fractions, Wfs) in matching target and reproduced numerosity as a function of target number. Average Wfs (across target number) in the fast and slow conditions were virtually identical (average = 0.17, SD = 0.01 and average = 0.17, SD = 0.007 for fast and slow tapping respectively) and within each condition, Wfs were roughly constant across targets, following Weber's Law.

We then computed the correlations of Wfs across participants, for all target values (see Methods for details). *Figure 3A, C* show correlation matrices across all participants (like the simulations of

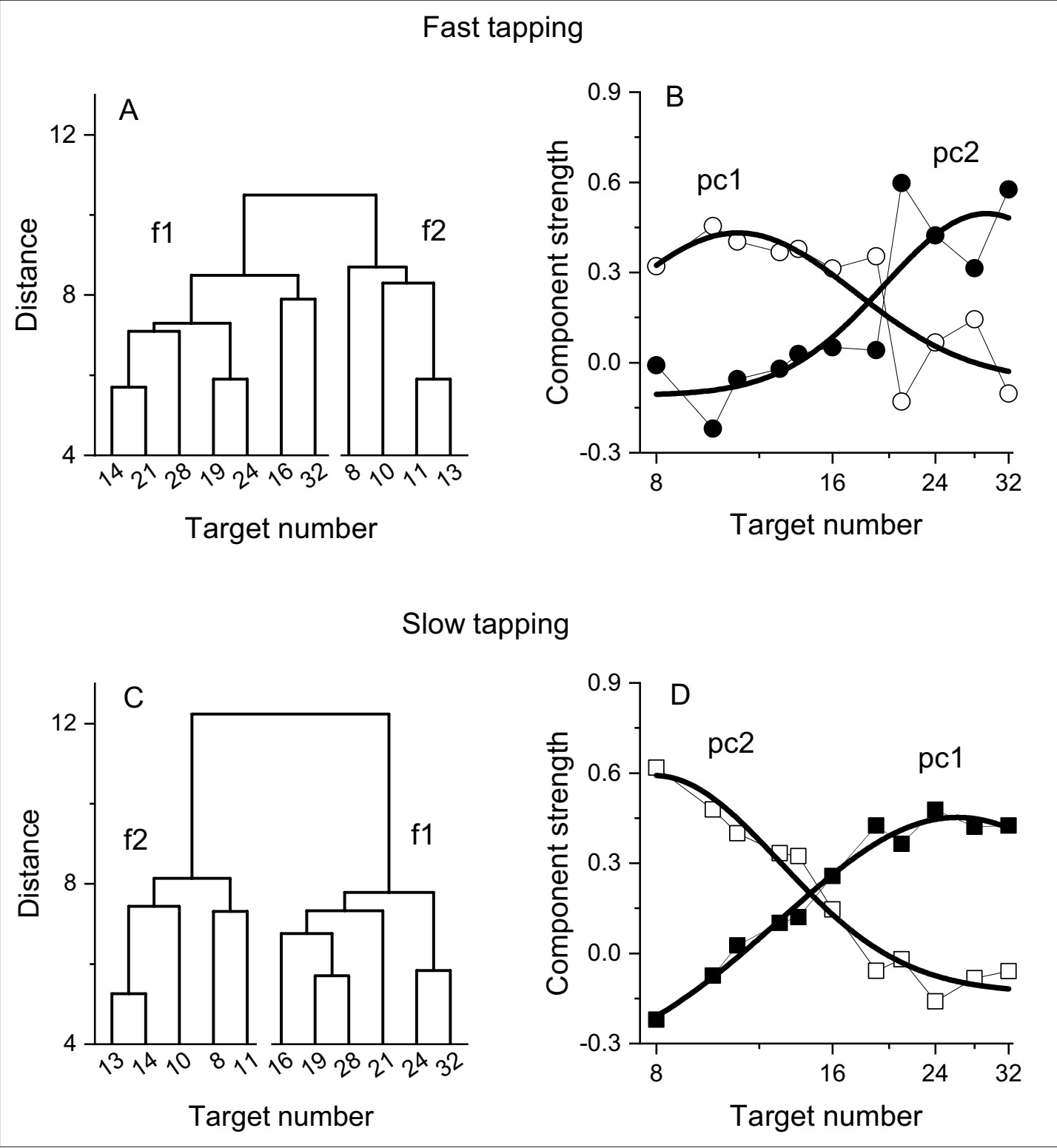

**Figure 4.** Dendrograms from hierarchical clustering on Wfs in the fast (**A**) and slow (**C**) tapping conditions. Panels B and D show the strength of the first two principal components from a PCA on the Wfs measured in the fast (**B**) and slow (**D**) tapping conditions (N30 and N29 respectively). The strength of these components is shown as a function of the different numerical target levels. The smooth curves are logGaussian fits to the component strengths.

*Figure 1*), with each cell representing the Pearson correlation coefficient between the two target numbers. The correlations were all positive, ranging from 0.1 to 0.75 in the fast condition and 0.3–0.88 in the slow condition. Importantly, the correlations were clearly not randomly distributed, but most of the higher correlations (reddish) lie near the diagonal, indicating that the motor matching performance was more similar for neighboring numbers, compared to higher numerical distance.

*Figure 3B, D* show average correlation coefficients, averaged into six bins as a function of numerical distance. There is a systematic decreasing trend in both fast and slow tapping conditions (linear fit on unbinned data shown by dashed lines in *Figure 3B, D*: slopes = –0.58±0.11,–0.57±0.07 for fast and slow respectively, both p<0.001), clearly different to that predicted by a random pattern of matching precision levels (curves with black filled symbols in *Figure 3B, D*, see Methods). Overall, these analyses confirm that the degree of performance similarity was systematically modulated by the numerical difference between targets, in line with the existence of underlying tuning functions.

As a preliminary analysis, we modeled the data with simulations like that of *Figure 1*, assuming one, two or four numerosity-selective channels. The average WFs for the participants was taken from the data, and channel width, variability, and measurement noise free parameters. The fits with a single channel were very poor, with $R^2$=0.003 and 0.002 (essentially no better than the mean) for the fast and slow tapping respectively. However, the two-channel model gave much better fits, with $R^2$=0.54 and 0.29. The four-channel model had similarly good fits, with $R^2$=0.52 and 0.28. The parameters to yield best fits seemed reasonable, with channel width of 1.2 and 1.8oc taves, and channel variability between 0.051 and 0.106.

We then investigated the structure underlying the number sensorimotor translation system with a hierarchical clustering analysis (see Methods for details). *Figure 4A, C* shows the resulting dendrograms. In both the fast and slow tapping conditions, the analyses returned a two-cluster solution with one cluster aggregating low numbers (N8-13 and N8-14 in the fast and slow conditions) and the other aggregating high numbers (N>13 and N>14, respectively). This analysis corroborates the results obtained from the correlation analysis and add on this suggests the existence (at least) of two

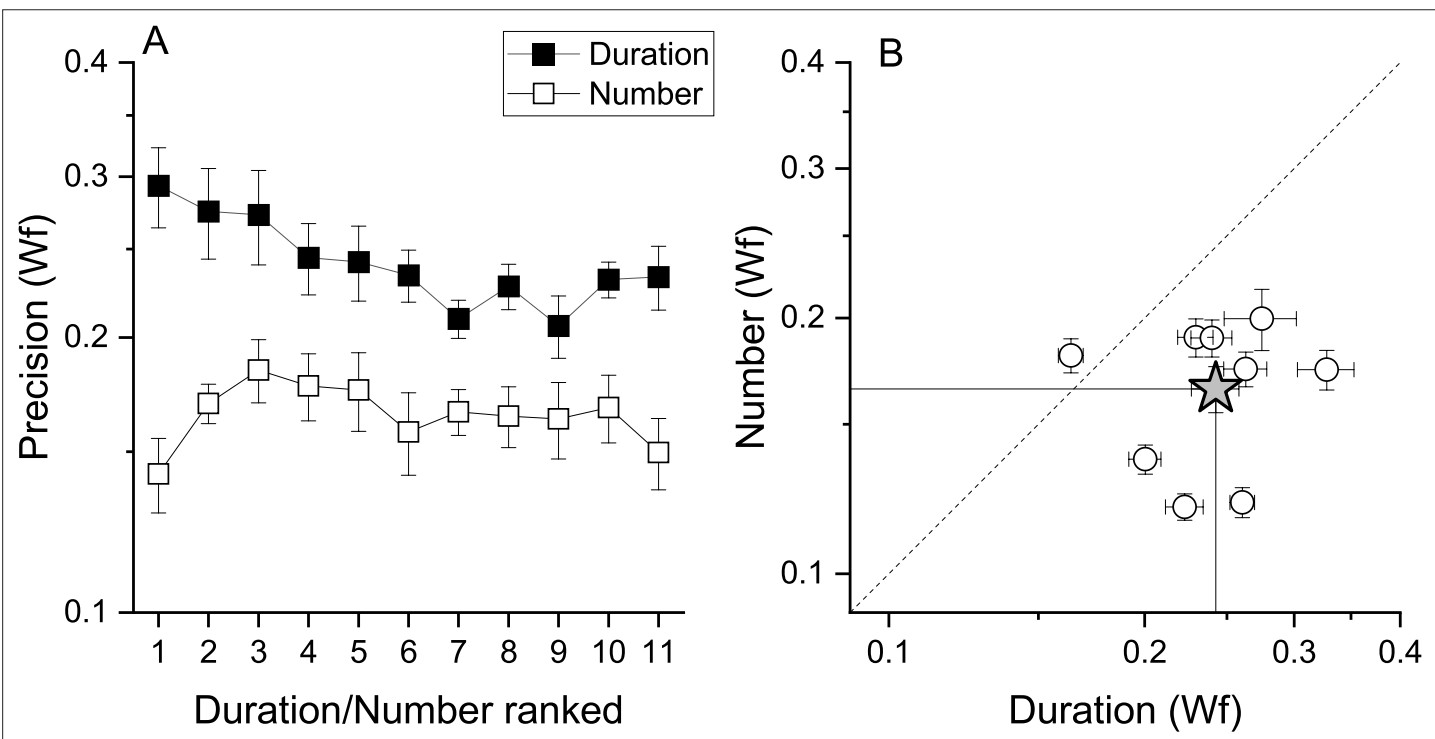

**Figure 5.** Number and duration matching tasks. (**A**) Average (across participant, N9) Weber fraction for the number (filled squares) and duration (empty squares) matching tasks, as a function of ranked target stimulus (visual digits for the number task, and the corresponding auditory tones durations for the duration task, see Methods for details). (**B**) Individual Weber fractions for the number task plotted against those for the duration task, averaged across stimulus intensity. The star symbol reports average across participants. Error bars are±1s.e.m. Weber fractions in the duration task were higher compared to those measured in the number task (RM ANOVA, p=0.002).

channels dealing with relatively different numerical targets. A control analysis on randomised data (see Methods) yielded clusters containing contiguous numbers only in 0.005% of cases (for both the fast and slow tapping conditions), suggesting that the results with the original dataset likely represent a structure genuinely organised as a function of the highest similarity between neighboring numbers.

To study the tuning of these two hypothetical channels, we then performed a factor analysis on Wfs. In both fast and slow tapping conditions, two factors emerged (see Methods), explaining a total of 68% and 81% of the total variance (for the fast and slow tapping conditions, respectively). *Figure 4* shows the rotated component strength as a function of target number. The results revealed two bell-shaped tuning functions for both fast (panel B) and slow (panel D) tapping conditions, with factor strength distributions reasonably described by logGaussian functions (fast tapping: $R^2=0.6$ and $R^2=0.65$; slow tapping: $R^2=0.97$, $R^2=0.94$ for 1st and 2nd components). In the fast-tapping condition, the first component peaked at lower target numbers (N 11) and the second component at higher numbers (N 29). In the slow motor condition, the first component packed at higher target numbers (N 26) and the second component at lower numbers (N 8).

That fast and slow tapping conditions provided similar results despite the very different tapping rates suggests that temporal frequency of the tapping did not play a major role. However, as total response duration was positively correlated with target numbers ($r=0.95$, $r=0.36$, for fast and slow tapping, both $p<0.001$), participants might have used response duration instead of number of taps as a stop criterion. To test this possibility, a sub-sample of 9 participants were given a control task to match the duration of a tone (see Methods). If the number task was performed through duration, this latter must have an equal or lower precision level compared to that measured in the number task. *Figure 5A, B* shows that this was not the case with the precision level (Wfs) in the number task being clearly lower (higher precision) than the duration task, making the use of duration strategies during the number task unlikely (number: average = 0.16, SD = 0.03; duration: average = 0.24, SD = 0.04). A RM ANOVA with task (duration, number) and stimuli intensity (11 levels) confirmed the difference ($F(1, 80) = 19.97$, $p=0.002$).

## Discussion

Using a motor reproduction task, we measured the precision with which number digits were translated into sequences of actions. Replicating previous evidence from both human and animal studies (*Cordes et al., 2001*; *Kirschhock and Nieder, 2022*; *Whalen et al., 1999*), precision followed Weber's law, a well-established signature of the Approximate Number System (*Anobile et al., 2014*; *Dehaene, 2011*; *Feigenson et al., 2004*; *Ross, 2003*). As the focus of the current study was on the individual differences in the number-to-action transformation process, we used digits as sensory stimuli. Digits encoding, at least with educated adults, is noiseless, eliminating this source of variability but leaving that associated with the sensorimotor transformation process.

Importantly, by looking at the covariance structure of the motor reproduction precision index (Weber fraction) across participants, we found evidence for sensorimotor channels tuned to number. Precision for reproducing numerically similar targets was positively correlated ($r\sim0.8$) but for larger numerical distances, correlations decreased ($r\sim0.1$), suggesting the action of sensorimotor channels sharing the encoding of neighboring numbers. In line with this, cluster analyses identified two main clusters, one aggregating low numbers (from 8 to ~13) and the other higher numbers (from ~13 to 32), pointing to (at least) two mechanisms encoding low and higher numbers. Finally, Principal Component Analyses on Weber fractions confirmed this and went to describe two bell-shaped components, one peaking at relatively low numbers (~10) and the second at higher numbers (~27).

The covariance technique used here has been widely employed to reveal visual channels in several basic perceptual dimensions such as colour, motion, contrast and spatial frequency (*Morrone et al., 1999*; *Peterzell et al., 1995*; *Peterzell et al., 2000*; *Peterzell and Kennedy, 2016*; *Peterzell and Teller, 1996*; *Rammsayer and Troche, 2014*). The technique is based on the idea that interindividual variability conveys information that can reveal common sensory processes (*Peterzell and Kennedy, 2016*), as illustrated in *Figure 1*. In practice, this approach requires multiple and densely sampled measurements of different stimuli intensity, in the same sample. Regions of high intercorrelation between neighboring stimuli intensity can be interpreted to imply that sets of stimuli are processed by the same (shared) underlying channel. This channel, while responding relatively more to its preferred stimulus, will also be activated by neighboring stimuli that although slightly different

from the preferred intensity, are nevertheless included in the same response distribution. Following this rationale, relatively lower correlations among more distant stimuli would indicate that these are processed by different, or at least partially independent, channels (*Peterzell and Teller, 1996*). The results obtained in this study are in line with those predicted by this technique, implying the existence of sensorimotor channels tuned to different numerical targets, with a rather wide response distribution. These channels probably reflect the activity of noisy neurons responsible for translating sensory information into the corresponding numerical motor output, such as those observed in crows (*Kirschhock and Nieder, 2022*).

The results reinforce adaptation studies showing number-selective adaptation between actions and perceived numerosity (*Anobile et al., 2016*; *Maldonado Moscoso et al., 2020*). Those studies provided strong evidence for the existence of at least two (but possibly more) sensorimotor mechanisms, tuned to high and low numbers. The results are also broadly in line with recent evidence from animal physiology showing number selective sensorimotor neurons in the crow brain, translating visual inputs (digits and dot arrays) into number of pecks (*Kirschhock and Nieder, 2022*). While the crow study found as many channels as the number tested (N1–5), we found evidence for only two channels (clusters and factors) over the range of 8–32. However, it is important to note that PCA aims to summarize the dataset with the minimal number of components (channels). We can therefore not exclude the possible existence of more than two (perhaps not fully independent) channels. The results should be not interpreted as evidence for only two sensorimotor channels, but rather as evidence that tuned mechanisms exist in the human brain, with at least two different tunings. This is also apparent from the simulations of *Figure 1*, showing that the predictions of two or multiple channels are very similar, difficult to distinguish with the current technique. Indeed, the fits to the data using two or four channels were very similar, but both almost infinitely better than the one-channel fit. In other words, the channels described here likely represent a coarse summary representation of several (probably sharper) underlying channels.

The results reported here are unlikely to be contaminated by strategies capitalizing on duration of motor responses and/or temporal frequency. Despite the very different tapping rates in the slow (~3 Hz) and fast motor conditions (~7 Hz), the overall pattern of results remained almost unchanged. The only difference was an inversion of the order of the principal components and factors, with the first component peaking at lower numbers for the fast-tapping condition and higher numbers for the slow tapping condition (and vice versa for the second component). This result suggests that for fast actions lower number targets show more variance than higher numbers, and vice versa for slower actions. Although we have no definitive explanation for this pattern of results, it could arise from different effects of the matching motor noise associated with different tapping rates across the number ranges. In any event, the results showed two covariance channels for both conditions, similarly distributed along the numerical targets, supporting the existence of at least two mechanisms, one tuned to the lower numbers and one to the higher (at least within this numerical range).

Even if total duration of responses were positively correlated with target number, and although participants were free to use duration as a viable stopping criterion, the exploitation of this non-numerical strategy seems unlikely. Duration could conceivably be useful proxy for numerosity, but only if the thresholds for duration were lower than those for numerosity: but the results show they are in fact higher. Once again, these results are in line with previous studies showing lower sensitivities for motor matching duration tasks compared with motor number matching tasks. The electrophysiological study on crows also found similar neural response curves across very different motor response timing conditions (*Kirschhock and Nieder, 2022*). In summary, the results seem to suggest that the channels found here, like those in crows, reflect a genuine number selective sensorimotor transformation processes.

What may be the adaptive value of this mechanism? The spontaneous use of the number of actions in the animal kingdom is now well-established. For example, desert ants rely on the number of steps to return to the nest (*Wittlinger et al., 2006*), some species of male frogs match or exceed the chucks of competitors to attract female partner (*Rose, 2017*), and some songbirds modulate the numbers of syllables in their calls to signal dangerousness of predators (*Suzuki, 2016*; *Templeton et al., 2005*). In all cases, the use of number-related motor strategies by animals has a clear ecological value for survival and species preservation.

This system may have been preserved in humans and repurposed for our species-specific needs. Some hints come from the literature on the role of visual numerosity perception (dot arrays). Although most animal species are capable of some form of numerosity estimation (*Butterworth, 2022*), it appears that in humans numerosity estimation may act as a non-symbolic precursor for the development of symbolic mathematical skills (*Chen and Li, 2014*; *Decarli et al., 2023*; *Halberda et al., 2008*; *Mazzocco et al., 2011*; *Piazza, 2010*; *Piazza et al., 2010*; *Schneider et al., 2017*). The sensorimotor system could act as an early tool to constantly update and calibrate motor and sensory systems for the encoding and active manipulation of quantities and objects, with a potential impact on the quality of formal mathematical skills in later development. In line with this idea, there are reports of a possible co-occurrence of motor impairments in children with mathematical learning disorders such as dyscalculia (*Westendorp et al., 2011*), and children with developmental coordination disorders (which impair gross and fine motor function) perform poorly on math and numerosity tasks (*Gomez et al., 2015*; *Gomez et al., 2017*). This is in line with the suggestive idea proposed by *Walsh, 2003* of a sensory magnitude system, encoding abstract quantities such as space, time and number, to guide planning and execution of actions. It is also in line with more recent proposal suggesting that the acquisition of number meaning is deeply grounded in sensorimotor experiences (*Ranzini et al., 2022*; *Sixtus et al., 2023*). This is the first study showing sensorimotor channels for the transformation of numbers into action sequences in humans, and many questions remain open. Among these: is the channel preference (the peak) fixed on the absolute numerical value, or does actively recalibrate according to the numerical context (range)? Do these channels also integrate non-symbolic numerical information such as the numerosity of sets of elements and/or sequences of visual/acoustic events? Are these channels active during motor programming, or do they require action execution? Do these channels integrate non-numerical motor parameters, such as the associated effort? Are these channels motor effector-specific? These (and probably other) questions are waiting to be explored using this technique, which has the advantage of being sufficiently generalisable to be expanded to investigate most stimulus features.

## Acknowledgements

This research was funded by the European Union (EU) and Horizon 2020—Grant Agreement no. 832813—ERC Advanced "Spatio-temporal mechanisms of generative perception — GenPercept"; and from the Italian Ministry of Education, University, and Research under the PRIN2017 program (Grant no. 2017XBJN4F— "EnvironMag").

## Additional information

### Funding

| Funder | Grant reference number | Author |
| --- | --- | --- |
| Horizon 2020 Framework Programme | 832813 | David Burr |
| Ministero dell'Istruzione, dell'Università e della Ricerca | 2017XBJN4F | Giovanni Anobile |

The funders had no role in study design, data collection and interpretation, or the decision to submit the work for publication.

### Author contributions

Giovanni Anobile, Conceptualization, Data curation, Software, Formal analysis, Supervision, Investigation, Methodology, Writing - original draft, Writing - review and editing; Irene Petrizzo, Data curation, Software, Formal analysis, Investigation, Visualization; Daisy Paiardini, Data curation, Investigation; David Burr, Conceptualization, Formal analysis, Supervision, Funding acquisition, Methodology, Writing - original draft, Project administration, Writing - review and editing; Guido Marco Cicchini, Conceptualization, Software, Formal analysis, Supervision, Visualization, Methodology, Writing - original draft, Project administration, Writing - review and editing

## Author ORCIDs
Giovanni Anobile (iD) http://orcid.org/0000-0003-2796-0661
Irene Petrizzo (iD) http://orcid.org/0000-0001-8736-8278

## Ethics

The experimental procedures were approved by the local ethics committee (Commissione per l'Etica della Ricerca, University of Florence, July 7, 2020, n. 111). The research was in accordance with the Declaration of Helsinki and informed consent were obtained from all participants prior to the experiment.

Reviewer #1 (Public Review): https://doi.org/10.7554/eLife.92169.3.sa1
Reviewer #2 (Public Review): https://doi.org/10.7554/eLife.92169.3.sa2
Reviewer #3 (Public Review): https://doi.org/10.7554/eLife.92169.3.sa3
Author Response https://doi.org/10.7554/eLife.92169.3.sa4

---

# Additional files

## Supplementary files
• MDAR checklist

## Data availability

All data generated or analysed during this study are available at: https://doi.org/10.5281/zenodo.8279068.

The following dataset was generated:

| Author(s) | Year | Dataset title | Dataset URL | Database and Identifier |
|---|---|---|---|---|
| Anobile G, Petrizzo I, Paiardini D, Burr DC, Cicchini GM | 2024 | Sensorimotor mechanisms selective to numerosity: evidence from individual differences | https://doi.org/10.5281/zenodo.8279068 | Zenodo, 10.5281/zenodo.8279068 |

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
