## [Editor Report · eLife assessment]

This potentially **important** paper addresses the question of how numerical information is represented in the human brain. Experimental findings are interpreted as providing evidence for a sensorimotor mechanism that involves channels, each tuned to a particular numerical range. While this is an interesting application of methodologies used to identify the presence of channels, the evidence supporting the claim that these have a sensorimotor basis is **incomplete**.

---

## [Referee Report · Reviewer #1 (Public Review)]

Anobile and colleagues present a manuscript detailing an account of numerosity processing with an appeal to a two-channel model. Specifically, the authors propose that the perception of numerosity relies on (at least) two distinct channels for small and large numerosities, which should be evident in subject reports of perceived numerosity. To do this, the authors had subjects reproduce visual dot arrays of numerosities ranging from 8 to 32 dots, by having subjects repetitively press a response key at a pre-instructed rate (fast or slow) until the number of presses equaled the number of perceived dots. The subjects performed the task remarkably well, yet with a general bias to overestimate the number of presented dots. Further, no difference was observed in the precision of responses across numerosities, providing evidence for a scalar system. No differences between fast and slow tapping were observed. For behavioral analysis, the authors examined correlations between the Weber fractions for all presented numerosities. Here, it was found that the precision at each numerosity was similar to that at neighboring numerosities, but less similar to more distant ones. The authors then went on to conduct PCA and clustering analyses on the weber fractions, finding that the first two components exhibited an interaction with the presented numerosity, such that each were dominant at distinct lower and upper ranges and further well-fit by a log-Gaussian model consistent with the channel explanation proposed at the beginning.

Overall, the authors provide compelling evidence for a two-channel system supporting numerosity processing that is instantiated in sensorimotor processes. A strength of the presented work is the principled approach the authors took to identify mechanisms, as well as the controls put in place to ensure adequate data for analysis.

One remaining question regards the secondary timing task that was used as a control. There may be interesting findings here to pursue, and so I encourage the authors or other researchers to examine those findings and explore further studies there as well.

---

## [Referee Report · Reviewer #2 (Public Review)]

Summary:

The authors wish to apply established psychophysical methods to the study of numbers. Specifically, they wish to test the hypothesis - supported by their previous work - that human sensorimotor processes are tuned to specific number ranges. In a novel set of tasks, they ask participants to tap a button N times (either fast or slow), where N varies between 8 and 32 across trials. As I understood it, they then computed the Weber fraction (WF) for each participant for each number and correlated those values across participants and numbers. They find stronger correlations for nearby numbers than for distant numbers and interpret this as evidence of sensorimotor tuning functions. Two other analyses - cluster analyses and principal component analyses (PCA) - suggest that participants' performance relied on at least 2 mechanisms, one for encoding low numbers of taps (around 10) and another encoding larger numbers (around 27).

Strengths:

Individual differences can be a rich source of scientific insight and I applaud the authors for taking them seriously.

Weaknesses:

Implications of intercorrelation. The experiment "is based on the idea that interindividual variability conveys information that can reveal common sensory processes (Peterzell & Kennedy, 2016)" but I struggle to understand the logic of this technique. The authors explain it most clearly when they write "Regions of high intercorrelation between neighbouring stimuli intensity can be interpreted to imply that sets of stimuli are processed by the same (shared) underlying channel. This channel, while responding relatively more to its preferred stimulus, will also be activated by neighbouring stimuli that although slightly different from the preferred intensity, are nevertheless included in the same response distribution." Why does high intercorrelation imply a shared channel and why should it be calculated across participants? Shouldn't performance on any set of tasks (that vary in difficulty) correlate across participants? Why in principle should people have distinct channels for processing similar stimuli and how could such a system improve (rather than impede) discrimination abilities? What pattern of intercorrelation would disconfirm the existence of tuning mechanisms? And perhaps most fundamentally: What is a channel and why do they matter?

Different channels? I had trouble understanding much of the analyses, and this may account for at least some of my confusion. That said, as I understand it, the results are meant to provide "evidence that tuned mechanisms exist in the human brain, with at least two different tunings" because of the results of the clustering analysis and PCA. But as the authors acknowledge, "PCA aims to summarize the dataset with the minimal number of components (channels). We can therefore not exclude the possible existence of more than two (perhaps not fully independent) channels." I would go a step further and say this technique does not provide more evidence for the existence of 2 channels as for the existence of 4, 8 or 24 channels, the upper bound for a task testing 24 different numbers. If we can conclude that people may have one channel per number, what does "channel" mean?

Several other questions arise when thinking through this technique, which left me skeptical of its utility. If people did have two channels (at least in this range), why would they be so broad? Why would they be centered so near the ends of the tested range? Can such effects be explained by binning on the part of the participants, who might have categorized each number (knowingly or not) as either "small" or "large"? Or by the kind of data-binning or distributions (i.e. Gaussian) used in the analyses? Or by the physical limits and affordances of the effector participants used (i.e. their finger)? Moreover, if people had sensorimotor channels tuned to different numbers, wouldn't this cause discontinuities in their own WF? Why look at correlations across individuals rather than correlations or discontinuities within individuals? Whereas the experiment tested numbers 8-32, numbers are infinite - How could a small number of channels cover an infinite set? Or even the set 8-10,000? What would the existence of multiple such channels mean for our understanding of numerical cognition? There may be good answers to these questions, but they are not clear to this reader.

Theories of numerical cognition. An expansive literature on numerical cognition suggests that many animals, human children, and adults across cultures have two systems for representing numerosity without counting - one that can represent the exact cardinality of sets smaller than about 4 and another that represents the approximate number of larger sets. Recent accounts suggest that what appears to be two systems can be explained by a single system of numerical approximation with limited information capacity (see Cheyette & Piantadosi, 2020). The current paper would benefit from better relating its findings to this long lineage of theories and findings in numerical approximation across cultures, ages, and species.

Specific to numbers? The authors argue that their effects are "number selective" but they do not provide compelling evidence for this selectivity. In principle, their main findings could be explained by the duration of tapping rather than the number of taps. They argue this is unlikely for two reasons. The first reason is that the overall pattern of results was unchanged across the fast and slow tapping conditions, but differences in duration were confounded with numerosity in both conditions, so the comparison is uninformative. The second reason is that temporal reproduction was less precise in their control condition than numerical reproduction, but this logic is unclear: Participants could still use duration (or some combination of speed and duration) as a helpful cue to numerosity, even if their duration reproductions were imperfect.

If the authors wish to test the role of duration, they might consider applying the same analytical techniques they use for number to their duration data. Perhaps participants show similar evidence for duration-selective channels, in the absence of number, as they do for other non-numerical domains (like spatial frequency).

---

## [Referee Report · Reviewer #3 (Public Review)]

Reviewing Editor's Summary:

The revised manuscript has clarified concerns raised by the reviewers concerning the analysis method in constructing the correlation matrix. These key results are now readily comprehensible. They have also added a final section to the Discussion, sketching some important questions for future research (e.g., number/resolution of channels and extension of the logic used here to look at number channels in other tasks).

Reviewer 1 was satisfied with these changes and has updated their review. Reviewer 2 did not think the revision tackled the theoretical issues raised in their initial review; as such, this reviewer has opted to leave their initial public review unchanged.

I also believe that the revision does not adequately address a major theoretical issue, namely whether the current data provide evidence of sensorimotor number channels, the central claim of the paper. The authors argue that since perception is noise free (stimuli were given symbolically), then the task variance comes from processes associated with sensorimotor transformation. Let's consider the task: A number is presented, the participant then attempts to produce that number of taps. To preclude counting, they are required to say the syllable "ba" as fast as possible while tapping. The sensorimotor channel idea would suppose that the symbolic stimulus activates a set of channels, each of which specifies the number of taps that should be produced. For example, a "6" channel likes to produce 6 outputs (with variability), a "10" channel 10 outputs (with variability), etc., with the actual production of the (weighted) integration of these outputs.

An alternative is that, since explicit counting is prevented by the secondary task, the participant makes an internal estimation of the number of produced taps. These judgments could be based on the output of amodal number channels. For example, the same process would be at play if the task were changed such that the participants watched a dot flash and had to estimate the number of flashes (while concurrently saying "ba"). The authors indicate in their response letter that they are conducting experiments along these lines and that the results are similar. They suggest that this provides support for the existence of both sensory and sensorimotor number channels. Extending this, if the experiment were tones instead of flashes, the argument would be that there are auditory, visual, and sensorimotor number channels. It seems more parsimonious to interpret such a pattern as reflective of amodal number channels.

I recognize there are other intriguing reasons to think there may be intimate links between our sense of number and movement, but I remain unconvinced that the current results provide evidence for sensorimotor number channels.

---

## [Author Response]

The following is the authors’ response to the original reviews.

**eLife assessment**
The paper addresses the important question of how numerical information is represented in the human brain. Experimental findings are interpreted as providing evidence for a sensorimotor mechanism that involves channels, each tuned to a particular numerical range. However, the logic of the channel concept as employed here, as well as the claims regarding a sensorimotor basis for these channels, is incomplete and thus requires clarification and/or modification.
**Reviewer #1 Public Review**
Anobile and colleagues present a manuscript detailing an account of numerosity processing with an appeal to a two-channel model. Specifically, the authors propose that the perception of numerosity relies on (at least) two distinct channels for small and large numerosities, which should be evident in subject reports of perceived numerosity. To do this, the authors had subjects reproduce visual dot arrays of numerosities ranging from 8 to 32 dots, by having subjects repetitively press a response key at a pre-instructed rate (fast or slow) until the number of presses equaled the number of perceived dots. The subjects performed the task remarkably well, yet with a general bias to overestimate the number of presented dots. Further, no difference was observed in the precision of responses across numerosities, providing evidence for a scalar system. No differences between fast and slow tapping were observed. For behavioral analysis, the authors examined correlations between the Weber fractions for all presented numerosities. Here, it was found that the precision at each numerosity was similar to that at neighboring numerosities, but less similar to more distant ones. The authors then went on to conduct PCA and clustering analyses on the weber fractions, finding that the first two components exhibited an interaction with the presented numerosity, such that each was dominant at distinct lower and upper ranges and further well-fit by a log-Gaussian model consistent with the channel explanation proposed at the beginning.Overall, the authors provide compelling evidence for a two-channel system supporting numerosity processing that is instantiated in sensorimotor processes. A strength of the presented work is the principled approach the authors took to identify mechanisms, as well as the controls put in place to ensure adequate data for analysis. Some questions do remain in the data, and there are aspects of the presentation that could be adjusted.The use of a binary colormap for the correlation matrix seems unnecessary. Binary colormaps between two opposing colors (with white in the middle) are best for results spanning positive and negative values (say, correlation values between -1 and +1), but the correlations here are all positive, so a uniform colormap should be applied. I can appreciate that the authors were trying to emphasize that a 2+ channel system would lead to lower correlations at larger ratios, but that's emphasized better in the numerical ratio line plots.

We agree and now changed the colour maps accordingly (Fig 1 and 3, p. 4 and 11). Thank you.

In Figure 1, the correlation matrices in Figure 1 appear blurred out. I am not sure if this was intentional but suspect it was not, and so they should appear like those presented in Figure 3.

Sorry about that, it was a rendering problem. Now fixed.

It's notable that the authors also collected data on a timing task to rule out a duration-based strategy in the numerosity task. If possible, it would be great to have the author also conduct the rest of the analyses on the duration task as well; that is, to look at WF correlation matrices/ratios as well as PCA. There is evidence that duration processing is also distinctly sensorimotor, and may also rely on similar channels. Evidence either for or against this would likely be of great interest.

We agree that investigating the existence of temporal channels would be of great interest, but it is goes beyond the scope of the current study. Out of curiosity, however, we analysed the duration data. Interestingly, signatures of sensorimotor channels (correlation gradient as a function on duration distance) emerge. Interestingly, this does not hold when correlating number against duration data. These results (if confirmed) would indicate the existence of independent mechanisms for the time and numerosity perception. Our research agenda is now proceeding in this direction.

For the duration task, there was no fast-tapping condition. Why not? Was this to keep the overall task length short?

Yes, this was the main reason.

The number of subjects/trials seems a bit odd. Why did some subjects perform both and not others? The targets say they were presented "between 25 and 30 times", but why was this variable at all?

The two experimental conditions were demanding, lasting around 2 hours each. Some participants, unfortunately, were available for just one slot. To make the two conditions similarly powered, we added some extra non-shared participants. Trials were divided into blocks of 55 trials (5 repetitions for each target). Most of the participants performed 6 blocks in both conditions, few of them (again for availability limits) performed 5 blocks.

For the PCA analysis, my read of the methods and results is that this was done on all the data, across subjects. If the data were run on individual subjects and the resulting PCA components averaged, would the same results be found?

We thank the reviewer for giving us the opportunity to clarify the technique.

In brief: we measured precision (Weber Fraction) in translating digits (target numbers) into corresponding action sequences. This creates a m by n matrix, each column (n) representing a participant, each row (m) a target number. This matrix was then submitted to PCA. The analyses provided two components. Each target number was assigned with two loading scores: one representing the loading on the 1st and one on the 2nd component. These loadings were than displayed as a function of targets, to describe the tunings. This analysis, by its nature, is across-participants and cannot be performed on individual data.

For the data presented in Figure 2, it would be helpful to also see individual subject data underlaid on the plots to get a sense of individual differences. For the reproduced number, these will likely be clustered together given how small the error bars are, but for the WF data it may show how consistently "flat" the data are. Indeed, in other magnitude reproduction tasks, it is not uncommon to see the WF decrease as a function of target magnitude (or even increase). It may be possible that the reason for the observed findings is that some subjects get more variable (higher WFs) with larger target numbers and others get less variable (lower WFs).

We agree and now added individual data, confirming flat WF distributions (Fig 2 B&D).

Regarding the two-channel model, I wonder how much the results would translate to different ranges of numerosities? For example, are the two channels supported here specific to these ranges of low and high numbers, or would there be a re-mapping to a higher range (say, 32 to 64 dots) or to a narrower range (say 16 to 32 dots). It would be helpful to know if there is any evidence for this kind of remapping.

This is the first study measuring sensorimotor channels for the transformation of numbers into action sequences. Whether these channels are modulated by the numerical context is an interesting open question that we are exploring through specific experimental conditions (now discussed at p. 17, lines 451-460).

**Reviewer #2 Public Review**
The authors wish to apply established psychophysical methods to the study of number. Specifically, they wish to test the hypothesis - supported by their previous work - that human sensorimotor processes are tuned to specific number ranges. In a novel set of tasks, they ask participants to tap a button N times (either fast or slow), where N varies between 8 and 32 across trials. As I understood it, they then computed the Weber fraction (WF) for each participant for each number and correlated those values across participants and numbers. They find stronger correlations for nearby numbers than for distant numbers and interpret this as evidence of sensorimotor tuning functions. Two other analyses - cluster analyses and principal component analyses (PCA) - suggest that participants' performance relied on at least 2 mechanisms, one for encoding low numbers of taps (around 10) and another for encoding larger numbers (around 27).StrengthsIndividual differences can be a rich source of scientific insight and I applaud the authors for taking them seriously, and for exploring new avenues in the study of numerical cognition.WeaknessesInter-subject-correlationThe experiment "is based on the idea that interindividual variability conveys information that can reveal common sensory processes (Peterzell & Kennedy, 2016)" but I struggled to understand the logic of this technique. The authors explain it most clearly when they write "Regions of high intercorrelation between neighbouring stimuli intensity can be interpreted to imply that sets of stimuli are processed by the same (shared) underlying channel. This channel, while responding relatively more to its preferred stimulus, will also be activated by neighbouring stimuli that although slightly different from the preferred intensity, are nevertheless included in the same response distribution." As I understood it, the correlations are performed "between participants, for all targets values" - meaning that they are measuring the extent to which different participants' WFs vary together. But why is this a good measure of channels? This analysis seems to assume that if people have channels for numerical estimation, they will have the same channels, tuned to the same numerical ranges. But this is an empirical question - individual participants could have wildly different channels, and perhaps different numbers of channels (even in the tested range). If they do, then this between-subject analysis would mask these individual differences (despite the subtitle).

Yes, the technique assumes that different individuals have similar channels, and the results confirm this. If everyone had different channels, or different numbers of channels, we would not have found this pattern of results: an ordered scaling of correlations as a function of numerical distance. As specified in the ms, however, this technique (at least as we used it) is not sensitive enough to identify the exact number of channels, so it may have smoothed the results, 'masking' the existence of more than two channels. To avoid possible confounds related to accuracy (reproduction biases), we used Weber Fraction, a standard index of normalized sensory precision (p. 7, lines 182-183).

Different channelsI had trouble understanding much of the analyses, and this may account for at least some of my confusion. That said, as I understand it, the results are meant to provide "evidence that tuned mechanisms exist in the human brain, with at least two different tunings" because of the results of the clustering analysis and PCA. However, as the authors acknowledge, "PCA aims to summarize the dataset with the minimal number of components (channels). We can therefore not exclude the possible existence of more than two (perhaps not fully independent) channels." So I believe this technique does not provide more evidence for the existence of 2 channels as for the existence of 4 or 8 or 11 channels, the upper bound for a task testing 11 different numbers. If we can conclude that people may have one channel per number, what does "channel" mean?

We recognise that the technique is not particularly intuitive, and we apologize for the lack of clarity.

To clarify: we measured the precision in translating digit numbers into action sequences. This was done for different target numbers (8, 10, 11, 13, 14, 16, 19, 21, 24, 28, 32) and with N participants. For each target number, and independently for each participant, we calculated the reproduction precision (Weber Fraction). The dataset comprised a matrix where each column represents a participant, and each row a target number. Each cell contains the corresponding Weber Fraction value. This dataset was then analysed with a simple correlation, across participants. For example, the WFs provided by the N participants when tested at the target number "8" were correlated with those obtained with the target number 10, 11, 13...32. The results show that the correlation between "8 and 10" (low numerical distance) was higher compared to that obtained correlating "8 with 32" (higher numerical distance). This pattern implies that the shared variance, between numbers, scales with numerical distance, across participants: implying the existence of channels aggregating similar numbers (i.e. tuning selectivity). On the same dataset we than ran a PCA. This analysis provides two main components. Within each component, each target number is assigned with a loading score: one for the 1st and one for the 2nd component. These loading were plotted as a function of targets, to describe the tunings shape (i.e. channels).

As stated above, we cannot really say exactly how many channels exist. These results should be interpreted as evidence for the existence of at least two channels for the transformation of numerical symbols into action sequences. This is not an obvious result at-all. There is no evidence in the literature for the existence of such mechanism in humans. In the animal (crow), there were found as many channels as the numbers tested. This does not contrast with our 2-channel results, but (very likely) arises from the different resolution of the techniques. Single cell recording has surely higher resolution compared to our interindividual covariance approach. In short, we believe that the channels revealed here are likely a coarse summary representation of several underlying channels.

We now tried to make these points clearer (p. 7 lines 186-196; p. 15 lines 382-384; p. 16 lines 401-402):

Several other questions arose for me when thinking through this technique. If people did have two channels (at least in this range), why would they be so broad? Why would they be centered so near the ends of the tested range? Can such effects be explained by binning on the part of the participants, who might have categorized each number (knowingly or not) as either "small" or "large"? Whereas the experiment tested numbers 8-32, numbers are infinite - How could a small number of channels cover an infinite set? Or even the set 8-10,000? More broadly, I was unsure what advantages channels would have - that is - how in principle would having distinct channels for processing similar stimuli improve (rather than impede) discrimination abilities?

This field of study is completely new, with many questions still open, including whether these channels are modulated by the numerical context such as the tested range and their extremes. The channels appear broad because, as stated above, they likely represent a coarse summary representation of several (probably sharper) underlying channels. We are now exploring the effect of numerical range and trying to modulate the tuning widths through ad-hoc experimental conditions. (p. 16 lines 401-402; p. 17 lines 450-459)

No number perceptionI was uncertain about the analogy to studies of other continuous dimensions like spatial frequency, motion, and color. In those studies, participants view images with different spatial frequency, motion, or color - the analogy would be to see dot arrays containing different numbers of dots. Instead, here participants read written numerals (like "19"), symbols which themselves do not have any numerical properties to perceive. How does that difference change the interpretation of the effects? One disadvantage of using numerals is that they introduce a clear discontinuity: Our base-10 numerical system artificially chunks integers into decades, potentially causing category-boundary effects in people's reproductions.

We used these sensory analogies to provide a flavour of the technique. The focus of the current study was on the individual differences in the numbers-to-actions transformation process. To this aim we decided to reduce the noise associated with the encoding of the sensory stimulus di per se. Digits encoding, at least with educated adults, is indeed noiseless, eliminating this source of variability. However, we agree that looking at non-symbolic formats would be interesting. We are now collecting data with dots and flash estimations. The results (so far) are largely in line with those found here, ensuring no chunking strategies, and confirming previous literature showing sensory numerosity selective channels in humans and animals. (p. 14 lines 351-355)

SensorimotorThe authors wished to test for "sensorimotor mechanisms selective to numerosity" but it's not clear what makes their effects sensorimotor (or selective to numerosity, see below). It's true they found effects using a tapping task (which like all behaviour is sensorimotor), but it's not clear that this effect is specific to sensorimotor number reproduction. They might find similar effects for numerical comparison or estimation tasks. Such findings would suggest the effect may be a general feature of numerical cognition across modalities.

Related to the above comment, the task here was to transform noiseless symbols (digits) into (noisy) numerical action sequences. Given that the source of variability is thus mainly driven by the sensory-to-action process, we believe that the task can be safely assumed to be considered sensorimotor in nature. (p. 14 lines 351-355)

Yes, the same pattern of results might be found for numerical comparison or estimation tasks, but using non-symbolic formats (dots/flashes). Educated adults make no errors in naming or comparing such simple digits, making this covariance analysis impossible to be performed with digit verbal estimation or comparison tasks. However, to anticipate our future results, we have preliminary data for dots and flashes verbal estimation tasks (“how many?”). The data are suggesting similar results, consolidating the technique, and confirming the large literature showing sensory channels for purely visual numerosity. (p. 17 lines 453-455)

Specific to numbersThe authors argue that their effects are "number selective" but they do not provide compelling evidence for this selectivity. In principle, their main findings could be explained by the duration of tapping rather than the number of taps. They argue this is unlikely for two reasons. The first reason is that the overall pattern of results was unchanged across the fast and slow tapping conditions, but differences in duration were confounded with numerosity in both conditions, so the comparison is uninformative. (Given this, I am not sure what we stand to learn by comparing the two tapping speeds.) The second reason is that temporal reproduction was less precise in their control condition than numerical reproduction, but this logic is unclear: Participants could still use duration (or some combination of speed and duration) as a helpful cue to numerosity, even if their duration reproductions were imperfect.If the authors wish to test the role of duration, they might consider applying the same analytical techniques they use for numbers to their duration data. Perhaps participants show similar evidence for duration-selective channels, in the absence of number, as they do for other non-numerical domains (like spatial frequency).

The fast and slow conditions were not meant to control for duration strategies but to test for the generalizability of the results over different tapping temporal dynamics (temporal frequency in this case). The results confirmed this.

The control for duration strategies is the comparison between precision in reproducing durations or numbers. In the number-to-action task, participants were free to use any cues, including response duration. However, it is safe to assume that the performance is dominated by the most precise feature, number in this case. In other words, in the number task if participants were reproducing the time required to give a certain number of presses, then in the timing task, where they are explicitly reproducing the same durations, they should show no lower precision. The results are opposite to that prediction. (p. 16 lines 418-420)

Theories of numerical cognition.An expansive literature on numerical cognition suggests that many animals, human children, and adults across cultures have two systems for representing numerosity without counting - one that can represent the exact cardinality of sets smaller than about 4 and another that represents the approximate number of larger sets (but see Cheyette & Piantadosi, 2020). The current paper would benefit from better relating its findings to this long lineage of theories and findings in numerical approximation across cultures, ages, and species.

The numbers used in this work were well above the subitizing limit (>N7). Indeed, the WFs found showed no signs of subitizing discontinuities. We believe that discussing the literature on subitizing here is too far from the scope of the current work.

Additional public comments from the Reviewing Editor:(1) What, in the present work, makes the case that the operative mechanism is sensorimotor? The authors frame the discussion around a sensorimotor number system but the evidence here could be seen as using a sensorimotor task as one way to get at an amodal number channel. For example, the authors could do the same experiment but have people watch a circle that flashes on and off for n times, with participants reporting the number of flashes (or shown a number and asked to say more or less). They could then apply the same analyses as used here. If they got the same results, it would seem that this would be an argument against the channels being sensorimotor. I suppose if they did NOT get results in the perceptual task, then they would have (much) stronger evidence that the channels are somehow sensorimotor in nature. Either way, an experiment along these lines would be essential for addressing the nature of the channels (tied to sensorimotor or not).

We chose to use this task because the perception of simple digits (like those used here), at least in educated adults, is noiseless. This ensures that the inter-individual variability remaining on the table is that related to the motor transformation process. For this reason, we believe that the task can be safely considered sensorimotor (see also Kirschhock & Nieder, Number selective sensorimotor neurons in the crow translate perceived numerosity into number of actions, Nature comm, 2022). (p. 14 lines 351-355)

This is not true for verbal numerosity estimation of non-symbolic stimuli (such as dots and/or series of events). It is well known that the estimation of the latter stimuli is noisy, and there would be no sensorimotor transformation processing in the task. The inter-individual variability in estimation precision and thus the measurable channels would then reflect sensory numerosity tunings. These have been revealed with various techniques in both humans and animals. However, we are now following this idea and we have preliminary data showing that sensory channels are also detectable by the technique used in the current study. This in not in contrast with the sensorimotor nature of the channels found here, but instead indicating the existence of both sensory and sensorimotor number channels.

The authors may argue that results from other studies such as the 2016 target article make the case about a sensorimotor basis of these channels. While I don't have a great grasp of this literature, my take on the 2016 target article is that the point was not about sensorimotor channels but about interactions between action and vision. This seems more in line with the idea of amodal number channels and indeed, they speak about a "generalized number sense" in that paper.

The 2016 paper showed that a short period of hand tapping (adaptation) can distort visual numerosity perception. The results implied the existence of sensorimotor number channels, integrating non-symbolic numerosity (dots/flashes) and actions. The current study goes beyond this, describing (for the first time) sensorimotor channels transforming symbolic numbers into action sequences. Whether these channels are also in charge to encode non-symbolic numerosity is an interesting open question that we are currently investigating with cross-tasks analyses. If the same channels are in charge to respond to non-symbolic numerosity (across space and time: dots and sequences of visual/auditory events) as well as to translate digits into actions, we could than speck about a generalized sensorimotor number sense. At present, this remains a possibility, to be tested. (p. 17 lines 450-459)

(2) There is a need for clarification on the method for creating the correlation matrices. The authors write that they look at correlations between Weber fractions between participants. By "between" do they mean "across"? That is, they calculate the Weber fraction for each individual for each cell. Then for a given cell, you correlate its Weber fraction with every other cell, using the pairs for each individual. I would call this "across" not "between." Is this just a semantic thing or have I misunderstood the process?To make this concrete, consider the correlation for cell 10/11. I assume it is something like10 11Subj1 .25 .31Subj2 .13 .09Subj3 .22 .16EtcAnd correlation across participants will be the data point for the 10/11 cell in the matrix.

It is a semantic error; this is exactly what we did: across participants.

To clarify better: we measured the precision in transforming numbers into sequences of actions. This was done for different target numbers (8, 10, 11, 13, 14, 16, 19, 21, 24, 28, 32) and with N participants. For each target number, and independently for each participant, we than calculated the reproduction precision (Weber Fraction). The dataset then consists of a matrix where each column represents a participant, and each row a target number. Each cell contains the corresponding Weber Fraction. This dataset was then analysed with a simple correlation, across participants. For example, the WFs of the N participants obtained when testing the target number "8" were correlated with those obtained with the target numbers "10, 11, 13...32". The results show that the correlation between "8 and 10" (low numerical distance) was higher compared to that obtained correlating "8 with 32" (higher numerical distance). This pattern implies that the shared variance, between numbers (across participants) scales with numerical distance, in line with the existence of channels that aggregate similar numbers (tunings).

(p. 7 lines 186-196)

(3) The duration data should be analysed. While n is small, can't the authors correlate WFs across tasks? Suppose a similar pattern is observed, suggestive of >1 channel in this between-task correlation.

One of the strengths of this technique is that it is very general, it can be applied to virtually every stimulus feature. We are currently collecting data to test the existence of generalised sensorimotor channels for continuous magnitudes: space, time, and numerosity. The logic is exactly as suggested. These correlational analyses however require (relatively) large samples and ad-hoc experimental conditions. We do not feel confident in providing messages on this with 9 participants. Out of curiosity, however, we analysed the data as requested and the results are interesting: signatures of sensorimotor channels emerge in both the number and duration tasks but NOT when analysed in conjunction (cross-task). If these results will be confirmed, would indicate the existence of separate mechanisms for the encoding of time and numerosity (and perhaps also space?).

(4) The finding of similar results for fast and slow is quite interesting. And provides good motivation to do the duration control experiment. But two issues related to the control experiment:(4a) Why not look at the correlation matrix for the duration task? Was this not done because there were only 9 participants? If so, why the small n here?

Yes, that is the reason. The aim of this work is not to investigate the existence of duration channels. This experimental condition was designed as a control for the use of non-numerical strategies in the number task. It worked well. The results were already obvious with 9 individuals (confirming Kirschhock & Nieder, Nature comm, 2022); we then did not consider necessary to continue in this direction. However, related to the previous point, we run a preliminary analysis on this small data set and (as mentioned above) signatures of sensorimotor channels (correlation gradients) emerge in both number and duration tasks but NOT when analysed in conjunction (cross-task), indicating different mechanism. We are now pursuing this issue using different number and duration tasks.

(4b) I don't follow why greater precision on the tapping task compared to the duration task makes a strong case against the duration hypothesis. Is the argument that, if based on duration, there should be greater precision on the duration task since the tapping task would exhibit the variability from duration PLUS added noise from tapping? If this is the argument, this should be spelled out.

Yes. The more precise feature dominates behaviour. In other words, in the number task if participants were reproducing the time required to give a certain number of presses, then in the timing task, where they are explicitly reproducing the same durations, they should show no lower precision. The results are opposite to that prediction. (p. 18 lines 418-420)

(4c) Related to point 3 above, one would expect based on things like Rammsayer's study that duration judgments would also engage channels. Is the idea that these are different channels in the tapping task? There seems a good case to have participants do both tapping and duration tasks and then do the correlation matrices, comparing within and between tasks.

Please see response to 3 and 4a.

**Recommendations for the authors:**
(1) On the logic of the channel concept as applied in the current context:While the authors present the numerical channel idea by analogy to how this concept is used for other features such as spatial frequency or orientation, there is no input to activate the channels-just a written numeral. The channel concept would mean that to respond to say, "16", you get output from multiple channels, with each weighted by its "tuning" to 16 such that the aggregate results in approximately 16 taps. This seems a bit odd: it would be like saying to draw, I use the output from my spatial frequency channels to create an image with a particular power spectrum. The logic of the channel concept in the current experimental context needs to be reviewed and clarified.

The channel here reflects (probably) the activity of noisy neurons in charge to translate sensory information into a numerical motor output, such as those shown by Kirschhock & Nieder (Nature comm, 2022) in the crows. We used digits because their encoding (at least for such simple digits and educated adults) has no associated noise. The interindividual variability left, and analysed, is thus mainly associated with the motor transformation process, revealing sensorimotor channels.

(2) A more thorough analysis of the duration task would strengthen the paper. The n is small for this interesting control condition and the analyses presented in the current version of the paper are limited. It is recommended to make this a fully powered test with complete analyses. Consider making this a new experiment in which participants do both the tapping and duration tasks to allow cross-modal analyses.

We run some exploratory analyses on this, described in comments 3 and 4a. We prefer to leave this issue to dedicated future experiments (which are just started).

(3) Expanded discussion of the limitations of the current study. The authors are clear that the methods don't provide a strong test of whether there are two or more than two channels. It would be useful to also comment on whether the estimated locations of the peaks are robust or if there is some sort of statistical bias for them to be at more extreme values. More generally, use the comments on the reviews to elaborate on various issues related to the channel concept.

We addressed these issues in the ms (p. 17 lines 450-459).

(4) Clarify the methods used to calculate the correlation matrix (see reviews).

We now specified better the correlation analyses (p. 7 lines 186-196).

(5) What is the basis for arguing that the mechanism under consideration is a "sensorimotor number system?" The data in this paper do not appear to provide evidence that the effects are linked to sensorimotor processes rather than reflect an amodal number system that is being accessed in their task through the motor system. At a minimum, present arguments for what motivates/justifies the sensorimotor claim or modify the paper to be neutral on this point.

We now specified better the sensorimotor nature of the task used here (p. 14 lines 351-355; see also comment 1).